# Effect of Subthalamic Nucleus Deep Brain Stimulation (STN-DBS) on balance performance in Parkinson's disease

**Haitao Li**[1]☯, **Siquan Liang**[1]☯*, **Yang Yu**[2], **Yue Wang**[2], **Yuanyuan Cheng**[2], **Hechao Yang**[3], **Xiaoguang Tong**[1]*

**1** Department of Neurosurgery, Tianjin Huanhu Hosptial, Tianjin, China, **2** Department of Neurological Rehabilitation, Tianjin Huanhu Hosptial, Tianjin, China, **3** Department of Psychology, Tianjin Huanhu Hosptial, Tianjin, China

☯ These authors contributed equally to this work.
* liangsiquan@163.com (SL); tongxghh@163.com (XT)

**Data Availability Statement:** There are ethical or legal restrictions on sharing a de-identified data set, because that the data contain potentially identifying or sensitive patient information and the ethics

## Abstract

### Purpose

To study the effect of STN-DBS on balance performance of Parkinson's disease.

### Method

16 idiopathic PD patients treated with bilateral STN-DBS (DBS Group) and 20 PD patients treated with Levodopa (Medicine group) were included in the study. Clinical material including Levodopa Equivalent Daily Dose (LEDD, mg/day), life quality (PDQ-39) were collected. For DBS group and Medicine group, The motor disability (Movement Disorder Society-Sponsored Revision of the Unified Parkinson's Disease Rating Scale , MDS-UPDRSIII) and balance performance (MDS-UPDRS 3.12, Berg Balance Scale BBS) and the Limits of Stability (LoS) (target acquisition percentage, trunk swing angle standard deviation, time) in state of Med-Off/Med-On at preoperation, postoperation, 6 months postoperation and 12 months postoperation were evaluated. Repeated ANOVA was used to analyze the effect of STN-DBS on balance performance.

### Result

The Clinical material (age, gender, duration, LEDD preoperation, PDQ39), motor disability (Med-on/Med-Off), balance performance (Med-on/Med-Off) and LoS preoperation had no differences in DBS-group and Medical-group ($P>0.05$). During the follow up, LEDD, PDQ39, Motor disability (MDS-UPDRSIII), balance performance (MDS-UPDRS 3.12, BBS) in Medicine-group had no significant changes in both Med-Off and Med-On. For DBS-group, immediately improvement of motor disability (MDS-UPDRSIII), LoS (target acquisition percentage, trunk swing angle standard deviation, time) and LEDD were observed postoperation ($P<0.05$); PDQ39, balance performance (MDS-UPDRS 3.12, BBS) began to improve at 6 months and 12 months postoperation. Repeated ANOVA showed that DBS could significantly improve the motor disability, balance performance and LoS in PD.

committee imposed them. Data requests may be sent to the Ethics Committee of Tianjin Huanhu Hospital (email:shhyyywk@tj.gov.cn).

**Funding:** The authors received no specific funding for this work.

**Competing interests:** The authors have declared that no competing interests exist.

## Conclusion

STN-DBS could improve the balance performance of PD patients in H&Y3.

## Introduction

Idiopathic Parkinson's disease (PD) is a common neurodegenerative disease in the elderly, with symptoms including bradykinesia, tremor, rigidity, and balance dysfunction. Balance dysfunction is one of the causes of posture instability. It decreases motor coordination as PD progresses. Balance dysfunction has been identified as the independent risk factors for falls. Approximately 45% to 68% of PD patients fall each year, with approximately two-thirds of them experiencing more than one fall [1]. Falling can have a series of serious consequences, including hip fractures and brain injuries. Due to the fear of falling, the activities of patients with PD are restricted, which seriously affects their quality of life [2]. Until now, improvement of balance function by different treatments remains ambiguous. How to better improve the balance function of patients is of great significance. In this study, we retrospectively analyzed patients to verify the effect of STN-DBS on balance performance.

## Methods

### Subjects

Sixteen idiopathic PD patients treated with bilateral STN-DBS (DBS Group) and 20 patients treated with Dopaminergic agents (Medicine-group, Med-group) at Tianjin Huanhu Hospital from December 2017 to August 2019 were retrospective analyzed in the study. Clinical characteristics, including gender, age, duration of PD, LEDD, and balance performance, were recorded and analyzed (Table 1). No information that could identify individual participants were included in this paper.

**Table 1. Clinical characteristics of patients in STN-DBS group and medicine group.**

| Characteristic | | DBS Group (n = 16) | Medicine Group (n = 20) | t/$\chi$2 | P |
|---|---|---|---|---|---|
| Age, years | | 60.25±5.56 | 57.88±6.98 | 1.33 | 0.26 |
| Gender, male | | 8,50% | 8,40% | 0.36 | 0.55 |
| Duration of PD, years | | 10.38±4.33 | 12.85±4.25 | 1.71 | 0.10 |
| LEDD, mg | | | | | |
| | preoperation | 1225.63±714.81 | 1200.80±714.81 | 0.11 | 0.91 |
| | 6 months | 503.13±140.50 | 1280.85±580.05 | 5.79 | 0.00* |
| | 12 months | 464.06±139.03 | 1270.55±688.28 | 4.86 | 0.00* |
| PDQ-39 | | | | | |
| | preoperation | 81.88±27.97 | 82.64±32.16 | 0.08 | 0.94 |
| | 6 months | 44.63±22.38 | 78.30±28.05 | 4.01 | 0.00* |
| | 12 months | 38.56±15.34 | 80.94±38.03 | 4.54 | 0.00* |
| MDS-UPDRS III preoperation | | | | | |
| | Med-off | 59.38±17.07 | 58.87±14.84 | 0.09 | 0.93 |
| | Med-on | 41.63±11.55 | 39.76±13.29 | 0.45 | 0.65 |

*P<0.01 means significant differences.

Inclusion criteria for DBS-group were: (1) idiopathic PD confirmed by 2015 MDS diagnostic criteria; (2) all the patients were in H&Y 3 stage.; (3) MDS-UPDRS III score improvement >30% in acute levodopa drug test before surgery; (4) qualified for surgical treatment and underwent bilateral STN-DBS (Chinese Parkinson's Disease Brain Deep Electrical Stimulation Therapy Expert Group, 2012). Exclusion criteria for DBS-group were: (1) balance dysfunction caused by secondary Parkinson's syndrome, hydrocephalus, intracranial tumors, and multiple cerebral infarctions. (2) patients with obvious cognitive dysfunction (Montreal Cognitive Assessment<26/30), serious anxiety, depression (Hamilton Anxiety Scale>21/56, Hamilton Depression Scale>24/68); (3) mental illness. All patients provided signed informed consent.

Inclusion criteria and Exclusion criteria for Med-group were as same as DBS-group except that patients refused DBS surgery for personal reasons.

## Ethical approval and registration

The study was approved by the Medical Ethics Committee of Tianjin Huanhu Hospital (No. 2019–35) and was conducted according to the principles of the Declaration of Helsinki. The study was registered with the China Clinical Trial Registration Center (ChiCTR1900022715).

## Surgical procedure

All surgeries were performed by the same surgical team. All patients underwent bilateral STN-DBS. The Leksell Stereotactic System (Elekta, Stockholm, Sweden) and Frame Link planning system (Medtronic, Minneapolis, Minnesota, USA) were used for surgery preparation. According to the Schaltenbrand-Wahren atlas, the tentative target site was 2 mm posterior to the midpoint of the anterior–posterior commissure (AC-PC) line, 12 mm lateral to the AC-PC line, and 4 mm ventral to the AC-PC line. Target sites were corrected based on T2-weighted MRI. The target was reconfirmed physiologically by an intraoperative microelectrode recording prior to performing the test stimulation studies. Quadripolar DBS electrodes (Activa 3389s, Medtronic) were implanted bilaterally as a stereotactic guide.

After inducing general anesthesia, implantable pulse generators (Activa RC Medtronic) were implanted subcutaneously in the subclavian pockets of the chest wall and connected to the DBS leads. Postoperative computed tomography images and preoperative MR images were superimposed in the Frame Link planning system to insure the local accuracy of electrode placement.

## DBS programming

All DBS device were switched on 4 weeks after the surgery (Med-Off). During programming, the side with more severe symptoms was programmed first. The corresponding contacts were tested one by one according to the patient's symptoms, with attention on the symptom control and stimulation parameters side effects. In principle, the best clinical symptom improvement is obtained with the minimum stimulation intensity [3].

## Medication

The initial drug dose postoperation is the same as preoperation. The drug dose was adjusted to be stable from 3 months to 6 months after surgery. During the follow up, all patients were underwent continuous rehabilitation training.

## Evaluation of quality of life, motor disability, balance performance, and limit of stability

The quality of life during follow-up was assessed by the PDQ-39. Motor disability (MDS-UPDRS III) and balance performance (MDS-UPDRS 3.12, BBS, LOS) were assessed at different times and status: preoperation (Med-off, Med-on), post-operation (4 weeks after operation, Stim-On/Med-Off, Stim-On/Med-On, the evaluate were underwent in 4 hours after the DBS turned on), 6 months postoperation (Stim-On/Med-Off, Stim-On/Med-On), and 12 months postoperation (Stim-On/Med-Off, Stim-On/Med-On).

Before the assessment, the patients stopped treatment with dopamine agonist for 72 hours, and compound levodopa and other anti-PD drugs for 12 hours.

During the test, the subject was not allowed to speak, look around, move upper limbs, or make other small movements. The subject was also told in advance what action was needed. After the instructing doctor issued relevant instructions, the subject performed the relevant action; if an action did not meet the requirements, the specified action was restarted.

## Statistical analysis

Statistical analyses were performed using SPSS 19.0 software. $t$-test or was used to analyze the quality of life, motor disability, balance performance, and limit of stability difference between STN-DBS group and Medicine group at the same time point. Repeated ANOVA was used to analyze the motor disability (MDS-UPDRSIII) and balance performance (MDS-UPDRS 3.12, BBS) and LoS)(target acquisition percentage, trunk swing angle standard deviation, time).

## Results

The mechanisms of action of DBS are still much debated. The authors should be less assertive.

We found no differences in the clinical characteristics of DBS Group and Medicine Group before the surgery (Table 1). The median age of the DBS Group was (60.25±5.56) years, with a median PD duration of (10.38±4.33) years. The LEDD preoperation in the DBS Group was (1225.63±714.81) mg, and PDQ-39 score was (81.88±27.97). The median age of the Med Group was (57.88±6.98), with a median PD duration of (12.85±4.25) years. The preoperation LEDD in the Med Group patients was (1200.80±714.81) mg, and PDQ-39 score was (82.64 ±32.16). LEDD and PDQ39 6month postoperation and 12month postoperation in DBS group were significant improved compared with Medicine group (Table 1). Motor disability, balance performance (Tables 2 and 3), and LoS (Tables 4 and 5) had no statistically significant differences preoperation in Med-Off and Med-On between the DBS Group and Medicine Group.

For the STN-DBS Group, immediate significant (P<0.05) improvement was measured in the Motor disability, target acquisition percentage, trunk swing angle standard deviation, and time as the DBS device switch on firstly postoperation. Balance performance (MDS-UPDRS 3.12, BBS) began to improve at 6 months postoperation. Repeated ANOVA showed that DBS could significantly improve motor disability, balance performance, and Los in PD. For the Medicine Group, Repeated ANOVA showed that motor disability, balance performance, and LoS were not improved during follow-up whether Med-on or Med-off (Tables 2–5).

During the follow up, MDS-UPDRS III score (Postoperation, 6 months postoperation, 12 months postoperation. Med Off),MDS-UPDRS 3.12 (6 months postoperation, 12 months postoperation. Med On/Med Off) and BBS (6 months postoperation, 12 months postoperation. Med On/Med Off) were better in DBS group. The Target acquisition (Postoperation, 6 months postoperation, 12 months postoperation. Med On/Med Off), Swing angle standard deviation (Postoperation, 6 months postoperation, 12 months postoperation. Med On/Med

**Table 2. Motor disability (MDS-UPDRS III) and balance performance (MDS-UPDRS 3.12, BBS) in states of Med-Off during follow-up.**

| | | Preoperation | Postoperation | 6 months | 12 months | | |
|---|---|---|---|---|---|---|---|
| | | Med-Off | Stim-On/Med-Off | Stim-On/Med-Off | Stim-On/Med-Off | F | P |
| MDS-UPDRS III | DBS Group | 59.38±17.07 | 39.94±7.74# | 36.63±6.83# | 31.44±4.72# | 14.28 | <0.01* |
| | Medicine Group | 58.87±14.84 | 56.89±10.25 | 52.79±10.28 | 52.06±10.24 | 3.94 | 0.05 |
| MDS-UPDRS 3.12 | DBS Group | 3.31±0.60 | 3.06±0.44 | 1.69±0.70# | 1.31±0.48# | 10.25 | <0.01* |
| | Medicine Group | 3.29±0.57 | 3.18±0.61 | 3.21±0.45 | 3.09±0.60 | 2.57 | 0.12 |
| BBS | DBS Group | 14.25±5.26 | 19.13±10.49 | 27.44±9.61# | 34.44±7.29# | 8.36 | <0.01* |
| | Medicine Group | 14.36±5.08 | 14.27±6.11 | 15.62±6.01 | 14.87±5.04 | 2.21 | 0.14 |

*:$P<0.01$ means significant differences.

#: $P<0.01$ compared with Medicine Group at the same time.

Off) and Time (6 months postoperation, 12 months postoperation. Med On/Med Off) was better in DBS group.

## Discussion

Factors affecting balance function in PD are complicated [4, 5]. With the application of DBS, the role of DBS on balance is controversy. As a first-line clinical drug, dopaminergic agents have a good therapeutic effect on bradykinesia, and rigidity. However, their effect on balance function are controversial [6]. This might be a result of the balance dysfunction in PD correlating with both the dopaminergic and cholinergic system [6]. Dopaminergic agents could effectively relieve dopamine deficiency symptoms while having no effect on the balance dysfunction related to the cholinergic system [7]. Moreover, due to pathological compensation with a long duration of bradykinesia, rigidity, and tremor, The body need to adapt new movement state when bradykinesia, rigidity, and tremor is controlled by dopaminergic agents. It would aggravate the balance dysfunction shortly.

In Medicine group, although no significant change was found in balance performance (MDS-UPDRS 3.12, BBS), LoS and PDQ39 during the follow up. A improvement trend of MDS-UPDRS III in Med On and LoS (swing angle SD and time) in Med Off. We speculated that it might be effect of continuous rehabilitation training or learning effect.

STN-DBS stimulation reduces the excitability of neurons in the STN, so as to suppress the hyperexcitability of SNr/GPi, which could inhibit the excitability of the SNr/GPi. This reduces excessive inhibition of the brain cortex [8, 9] and improve the symptoms of bradykinesia,

**Table 3. Motor disability (MDS-UPDRS III) and balance performance (MDS-UPDRS 3.12, BBS) in states of Med-On during follow-up.**

| | | Preoperation | Postoperation | 6 months | 12 months | | |
|---|---|---|---|---|---|---|---|
| | | Med-On | Stim-On/Med-On | Stim-On/Med-On | Stim-On/Med-On | F | P |
| MDS-UPDRS III | DBS Group | 41.63±11.55 | 39.31±7.06 | 35.44±6.21 | 30.25±3.99 | 13.91 | <0.01* |
| | Medicine Group | 39.76±13.29 | 38.80±10.68 | 37.99±11.25 | 37.61±10.22 | 3.79 | 0.07 |
| MDS-UPDRS 3.12 | DBS Group | 3.25±0.58 | 3.00±0.37 | 1.69±0.70# | 1.31±0.48# | 6.47 | <0.01* |
| | Medicine Group | 3.21±0.64 | 3.18±0.57 | 3.14±0.53 | 3.16±0.45 | 2.01 | 0.23 |
| BBS | DBS Group | 14.31±5.25 | 19.25±11.02 | 27.50±9.17# | 34.25±17.07# | 13.52 | <0.01* |
| | Medicine Group | 13.67±6.24 | 14.02±5.29 | 14.14±5.53 | 13.89±4.87 | 1.14 | 0.45 |

*$P<0.01$ means significant differences.

#: $P<0.01$ compared with Medicine Group at the same time.

**Table 4. Evaluation of LOS in states of Med-Off during follow-up.**

| | | Preoperation | Postoperation | 6 months postoperation | 12 months postoperation | | |
|---|---|---|---|---|---|---|---|
| | | Med-Off | Stim-On/Med-Off | Stim-On/Med-Off | Stim-On/Med-Off | F | P |
| Target acquisition, % | DBS Group | 20.79±6.62 | 60.97±10.10# | 62.59±9.18# | 71.80±7.01# | 7.25 | <0.01* |
| | Medicine Group | 21.87±7.03 | 24.73±13.27 | 23.04±14.82 | 22.74±12.06 | 2.20 | 0.18 |
| Swing angle standard deviation, ° | DBS Group | 0.62±0.11 | 0.30±0.07# | 0.22±0.05# | 0.18±0.04# | 9.05 | <0.01* |
| | Medicine Group | 0.71±0.13 | 0.63±0.13 | 0.65±0.17 | 0.62±0.13 | 3.25 | 0.09 |
| Time, s | DBS Group | 91.13±13.15 | 74.44±10.72 | 61.63±9.57# | 50.31±9.55# | 12.16 | <0.01* |
| | Medicine Group | 83.87±15.34 | 84.01±16.83 | 83.62±18.23 | 85.41±16.24 | 3.80 | 0.06 |

*$P<0.01$ means significant differences.

#: $P<0.01$ compared with Medicine Group at the same time.

rigidity; Otherwise, the STN and PPN are in close proximity, with close nerve fiber connections between nuclei. The PPN plays an important role in balance function, as it is an important part of MLR and has a close relationship with balance function [10, 11]. Some scholars have shown that stimulation of the PPN can alter motor behavior [12].

In DBS group, balance function was improved significantly. Motor disability and LoS immediately improved postoperation (4 weeks later postoperation when the device turn on firstly). Balance performance (MDS-UPDRS 3.12, BBS) began to improve 6 months and 12 months postoperation. In this study, the DBS Group had no significant changes in the balance-related scores immediately postoperation, and the stability limit of the patients was improved compared to preoperative. Therefore, the balance function in patients with DBS improved postoperation, while it was not accurately reflected due to the subjectivity and bias of the balance function score. Assessment forms such as UPDRS-III and BBS are commonly used to clinically evaluate the balance function, but has a great amount of bias due to the subjective judgment. The inaccuracy and non-objective assessment are obstacles to studying the pathophysiology of balance disorders. In this study, TecnoBody PROKIN system helped us in quantitative evaluate the balance function effectively.

Studies have pointed out that STN-DBS can improve the balance function in the case of sensory deprivation and sensory input inconsistency [13]. In addition to improving the bradykinesia, rigidity, and tremor symptoms of PD, STN-DBS may also improve the complications camptocormia, dyskinesia, and symptom fluctuations, which is helpful for improving balance function [14] while in other study, it tends to have no effect [15] that STN-DBS improves camptocormia, especially if surgery is performed more than 18 months after camptocormia

**Table 5. Evaluation of LOS in states of Med-On during follow-up.**

| | | Preoperation | Postoperation | 6 months postoperation | 12 months postoperation | | |
|---|---|---|---|---|---|---|---|
| | | Med-On | Stim-On/Med-On | Stim-On/Med-On | Stim-On/Med-On | F | P |
| Target acquisition, % | DBS Group | 56.90±6.38 | 65.63±9.71# | 66.65±8.18# | 74.32±5.86# | 13.77 | <0.01* |
| | Medicine Group | 55.08±7.03 | 53.24±10.86 | 50.33±14.23 | 51.35±12.85 | 1.28 | 0.48 |
| Swing angle standard deviation, ° | DBS Group | 0.43±0.10 | 0.29±0.07# | 0.22±0.04# | 0.18±0.05# | 10.04 | <0.01* |
| | Medicine Group | 0.42±0.13 | 0.42±0.15 | 0.41±0.20 | 0.40±0.17 | 2.98 | 0.10 |
| Time, s | DBS Group | 76.94±12.08 | 73.13±10.09 | 60.56±10.08# | 44.19±8.36# | 12.77 | <0.01* |
| | Medicine Group | 74.24±14.24 | 75.21±12.76 | 74.78±16.33 | 72.14±15.22 | 1.09 | 0.58 |

*$P<0.01$ means significant differences.

#: $P<0.01$ compared with Medicine Group at the same time.

installation [16]. And STN-DBS does improve balance in the short term (1–3 years) but that balance dysfunction then increases again with disease progression. Partly we thought that the different including criteria might lead to diverse outcomes. On the other hand, quantitative measures of balance could reveal the balance function objectively. . . After all, we found that STN-DBS could improve the short-term balance function in this study, though the long-term follow up and large sample size are needed for further study.

## Supporting information

**S1 Checklist.**
(DOCX)

**S2 Checklist.**
(DOCX)

**S1 File.**
(DOCX)

## Author Contributions

**Conceptualization:** Haitao Li, Siquan Liang, Hechao Yang, Xiaoguang Tong.

**Data curation:** Haitao Li, Xiaoguang Tong.

**Formal analysis:** Haitao Li.

**Funding acquisition:** Haitao Li.

**Investigation:** Haitao Li, Yang Yu.

**Methodology:** Haitao Li.

**Project administration:** Haitao Li, Yang Yu, Yue Wang.

**Resources:** Haitao Li, Yue Wang.

**Software:** Haitao Li.

**Supervision:** Haitao Li.

**Validation:** Siquan Liang, Yuanyuan Cheng.

**Visualization:** Yuanyuan Cheng.

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
