## [Decision Letter · Decision Letter 0]

8 Apr 2020

PONE-D-20-03478

Effect of subthalamic nucleus deep brain stimulation (STN-DBS) on balance performance in Parkinson's disease.

PLOS ONE

Dear Dr. Tong,

Thank you for submitting your manuscript to PLOS ONE. After careful consideration, we feel that it has merit but does not fully meet PLOS ONE’s publication criteria as it currently stands. Therefore, we invite you to submit a revised version of the manuscript that addresses the points raised during the review process.

We would appreciate receiving your revised manuscript by May 23 2020 11:59PM. To enhance the reproducibility of your results, we recommend that if applicable you deposit your laboratory protocols in protocols.io, where a protocol can be assigned its own identifier (DOI) such that it can be cited independently in the future. For instructions see: http://journals.plos.org/plosone/s/submission-guidelines#loc-laboratory-protocols

We look forward to receiving your revised manuscript.

Kind regards,

Karsten Witt

Academic Editor

PLOS ONE

Reviewers' comments:

Reviewer's Responses to Questions

**Comments to the Author**

1. Is the manuscript technically sound, and do the data support the conclusions?

Reviewer #1: Yes

Reviewer #2: Partly

2. Has the statistical analysis been performed appropriately and rigorously? 

Reviewer #1: I Don't Know

Reviewer #2: No

3. Have the authors made all data underlying the findings in their manuscript fully available?

Reviewer #1: Yes

Reviewer #2: No

4. Is the manuscript presented in an intelligible fashion and written in standard English?

Reviewer #1: Yes

Reviewer #2: No

5. Review Comments to the Author

Reviewer #1: Major comments.

1.According to the title, the authors seem to explore effect of STN-DBS on balance performance in Parkinson's disease. But the introduction of the paper focus on analyzing effect of DBS and medicine on balance performance.(page 2, line 44-45).

2.The description of statistical analysis was too brief to understand. For example, authors should have clarified which data nonparametric tests were used to analysis ? Was one-way repeated ANOVA used to analysis effect of STN-DBS or medicine on balance performance wrong writen? Whether it ought to be “repeated ANOVA” or not? And why only one “F” or “P” presented in Table 2-4. We think “F” value result from STN-DBS group or Med-group separately. That is, whether the results indicated that DBS could significantly improve motor disability, balance performance, and Los in PD with repeated ANOVA, which is the same as Med-group? In conclusion, we think it is necessary to describe the method of statistical analysis in detail.

3.The discussion was long-winded written and lack of logic. Page 6, line 147-157 discussed pathology of Parkinson’s disease and balance dysfunction, which was repeatedly writen in later paragraphs. Page 7, line 180-182: the sentence “ Studies have shown that APR abnormalities in patients with PD are related to decreased acetylcholinergic content in the thalamus, and are not related to acetylcholine in the cortex and striatum dopamine.” may be meaningless in this section. The paper intended to study effect of STN-DBS on balance performance in Parkinson's disease, so we think it’s unnecessary to analyze effect of PPN-DBS stimulation in Page 7, line 183-192. Moreover, we think discussion in the paper is lacking of focus. Maybe authors more ought to analyze the meaning of the result, and the differences with previous researches. Whether the result is consistent with previous studies, if not, what possible factors may lead to the differents? And are there some lightspots in the paper?

4.Some important thesis in the paper prefer to add some supporting literatures. Ex, page 8, line 202-203, STN-DBS can also improve the central axis symptoms by reducing the abnormal inhibition of PPN activity by the STN or compensating for neuronal loss in the PPN.

5.Authors seemed to only attributed the limitation of the study to the subjectivity of scale assessment. We think it may be unconscionable. As we know, this study was retrospective, which means the researcher could distinguish between the STN-DBS group and the Med group, leading to bias during assessing the scales. Moreover, there were also some confounding bias existing in the study deriving from retrospective study. In the other way, the sample size was small in the study, and it may lead to unstable result.

6.The writing of the paper need to improve. There were some sentences hard to understand.For example, page 6,line 161-162, ”This is consistent with a previous study [8] and may be a result of the balance dysfunction in PD correlating with both the dopaminergic and cholinergic system.”. Page 8,line 214-215, “Therefore, the balance function in patients with DBS improved postoperation but was not accurately reflected due to the subjectivity and bias of the balance function score.” and so on.

Minor comments

1.Page 2, line 41-42: Whether the sentence ”Due to the fear of falling, the activities of patients with PD are restricted, which seriously affects their quality of life, and even life safety” was wrong described? That restricting activities of patients with PD would seriously affects life safety?

2.Page 2, line 45: “different treatment” ought to be “different treatments” .

3.Page 4, line 89-90: “;” should be represented by “,”

4.Page 4, line 107: whether the “post-operation” represented “4 weeks after operation” or not?

5.Page 5, line 131: “We found no differences in the clinical characteristics of the DBS Group and Medicine Group” would be better to write as “We found no differences in clinical characteristics of the DBS Group and Medicine Group before operation.”

6.Page6, line 136: “The LEDD preoperation” ought to be “The LEDD in preoperation”

Reviewer #2: The authors performed a retrospective study of short term effects of STN-DBS stimulation on balance control. The issue is of importance in the field. A great strength of the study is that data from the surgery group are compared to data of a control, best medical treatment group. If correctly analyzed, such data would provide new and relevant information in the field. However, in the present version of the study report, there are major methodological weaknesses in data processing. In addition, the presentation of the methods and results needs to be thoroughly edited. The discussion needs to be focused on the reported data rather than on general considerations about the neuronal networks possibly involved in balance control and balance disorders in Parkinson’s disease patients. Overall, the paper needs serious editing and information needs to be presented in a more structured way. The abstract would also have to be rewritten accordingly.

Major concerns

* The specific variables recorded and analyzed need to be clearly presented and defined. Scoring needs to be explained. It is not clear when the first measure post-surgery was performed? (upon switching on stimulation a few days or weeks after surgery? At three months, after xx weeks of chronic stimulation and adjustments of parameter settings?)

* It is said in the text that non parametric tests were used, but also that one-way repeated ANOVA were used. Which ANOVA? Friedman? Kruskall-Wallis? GLM? As Tables report F values, it seems that parametric ANOVAs were used. The statistical design to be precisely described. For each dependent variable, there were 4 time-point measures (preop, sometime early post-op, 6 months follow-up and 12 months follow-up) X 2 medication conditions (med off, med on). The most appropriate design (using parametric statistic) would thus be a GLM or ANOVA with a design 2 conditions X 4 time-points (repeated measures for this last factor). If non parametric tests are used, the design is more complex and requires 2 steps.

* Demographical data cannot be analyzed as the dependent variables of the study. The two groups need to be compared using t tests or a non-parametric equivalent.

* Data are presented as mean and standard deviation in the Tables, and median and quartiles in the text for some of them (age, PD duration) but not all (LEDD). For the reader, it would be friendlier to have a coherent way of presenting the data. Statistical outcomes for the demographical data need to be reported. One would also want to see the MDS-UPDRS III total score Med off and Med on for the two groups, before surgery (there are given in Tables 2 and 3, but the reader would want to see them with the demographical data, as they contribute to describing the population).

* In Table 1, it is said that males represented 8.5 and 8.4% of the sample; this means that 92% of the patients were females? This would be very unusual and would need to be discussed. In addition, 8.5% of 16 patients are 1.36 persons, and 8.4% of 20 patients are 1.68 persons. This does not make much sense. Examination of Table 1, also shows that disease onset was about 50 years of age in average in the DBS group, and about 45 years of age in the medical treatment group. The latter is rather early onset. This should be discussed also.

* Regarding the dependent variables, Tables 2 and 3 report mean and SD, and a single F and p, without explanation of what these Fs and ps refer to: effect of group? Effect of time? What about interactions? Were they significant? If yes post-hoc analyses need to be reported.

In the text, the outcome of the statistical analyses needs to be precisely reported. “No obvious differences” (page 6, lines 138-139) is not an objective way of reporting. Either there were statistically significant differences, or there were not. Report successively the outcome for each variable.

Discussion

In its present form, the discussion does not discuss the results of the study. There are many general considerations on balance difficulties in PD, and potentially involved brain regions. The precise outcomes as well as characteristics of the sample and limitations of the study need to be specifically examined and interpreted.

Minor concerns

Introduction:

* what does “primary” PD means? Idiopathic? (page 2, line 35)

* What do you mean with “increases body swing” (page 2, line 37); please be specific in the description of the sign

* What does “decreases coordination” refer to? Which coordination are you referring to? (page 2, line 37)

* Aim of the study (page 2, line 44-45 “In this study, we retrospectively 45 analyzed patients treated by DBS and medicine to verify the effect of different treatment.”) is too unspecific and should be rephrased. Aim of the study seems to have been to compare the effect of medical treatment vs. STN-DBS on balance control in patients with PD.

Methods:

* Inclusion criteria: please make several sentences or a structured list to make the information legible (page 2 last line to page 3 line 57)

* What was the test and cut-off used to exclude patients with “obvious cognitive dysfunction”? (page 3, line 60); same issue for anxiety and depression.

DBS programming

* Page 4, line 85: contacts, not contactors

* Page 4, line 87: What does it mean to test the stimulation parameters “with attention on the … and drug side effects”? If testing the effect of the stimulation, the unwanted effects are related to stimulation parameters, not to drug.

* Page 4 lines 88-91. “Problems in balance….”. Delete this long sentence. This sentence refers to usual procedures of parameter are considerations on strategies that have been tried to improve balance or gait issues in patients with DBS. You may want to discuss these issues in the discussion, if some of the patients of the present study were actually stimulated in the SNr.

* Page 4 lines 98-103. Again these are general comments regarding management of patients after DBS implantation and is not really relevant to the present report. This paragraph should be deleted, except for the 1st and last sentences. (In addition, FOG can be induced by levodopa, so that increase in levodopa dose is not always the appropriate strategy for clinical management of FOG, especially in patients with dyskinesia.)

* Evaluation of quality of life, motor disability, balance performance and limit of stability (page 4 – lines 104 and following). This whole paragraph needs to be structured. Scales need to be presented (PDQ-39, item 3.12 of MDS-UPDRS III, BBS). Description of the LOS procedure needs editing to be fully understandable (for example, regarding the position of the patients, it would be simpler to state that patients were standing upright, feet at shoulder width, and looking straight ahead at the display. The task needs to be more clearly explained, as well as the variables (what is the “target body swing angle”?). There seems to be a contradiction between performing accurate movements to the target and maximizing movement. “Small limbs” are called “upper limbs” (I suppose).

6. PLOS authors have the option to publish the peer review history of their article (what does this mean?). If published, this will include your full peer review and any attached files.

Reviewer #1: No

Reviewer #2: No

---

## [Author Response · Author response to Decision Letter 0]

23 May 2020

Dear reviewers,

Thank you very much for giving us an opportunity to revise our manuscript. We appreciate the editor and reviewers very much for their constructive comments and suggestions on our manuscript entitled "Clinical experience of comprehensive treatment on the balance function of Parkinson's disease"(ID: MD-D-20-00182).

We have studied reviewers' comments carefully. According to the reviewer's detailed suggestions, we have made a careful revision on the orginal manuscript. All revised portions are marked in red in the revised manuscript. The corrections are as follows:

Reviewer #1: Major comments.

1.According to the title, the authors seem to explore effect of STN-DBS on balance performance in Parkinson's disease. But the introduction of the paper focus on analyzing effect of DBS and medicine on balance performance.(page 2, line 44-45).

Response:The introduction had been amendment carefully according to the comments.

2.The description of statistical analysis was too brief to understand. For example, authors should have clarified which data nonparametric tests were used to analysis ? Was one-way repeated ANOVA used to analysis effect of STN-DBS or medicine on balance performance wrong writen? Whether it ought to be “repeated ANOVA” or not? And why only one “F” or “P” presented in Table 2-4. We think “F” value result from STN-DBS group or Med-group separately. That is, whether the results indicated that DBS could significantly improve motor disability, balance performance, and Los in PD with repeated ANOVA, which is the same as Med-group? In conclusion, we think it is necessary to describe the method of statistical analysis in detail.

Response:The description of statistical analysis had been amendment carefully.

3.The discussion was long-winded written and lack of logic. Page 6, line 147-157 discussed pathology of Parkinson’s disease and balance dysfunction, which was repeatedly writen in later paragraphs. Page 7, line 180-182: the sentence “ Studies have shown that APR abnormalities in patients with PD are related to decreased acetylcholinergic content in the thalamus, and are not related to acetylcholine in the cortex and striatum dopamine.” may be meaningless in this section. The paper intended to study effect of STN-DBS on balance performance in Parkinson's disease, so we think it’s unnecessary to analyze effect of PPN-DBS stimulation in Page 7, line 183-192. Moreover, we think discussion in the paper is lacking of focus. Maybe authors more ought to analyze the meaning of the result, and the differences with previous researches. Whether the result is consistent with previous studies, if not, what possible factors may lead to the differents? And are there some lightspots in the paper?

4.Some important thesis in the paper prefer to add some supporting literatures. Ex, page 8, line 202-203, STN-DBS can also improve the central axis symptoms by reducing the abnormal inhibition of PPN activity by the STN or compensating for neuronal loss in the PPN.

Response:The discussion had been rewritten carefully according to the comments.

5.Authors seemed to only attributed the limitation of the study to the subjectivity of scale assessment. We think it may be unconscionable. As we know, this study was retrospective, which means the researcher could distinguish between the STN-DBS group and the Med group, leading to bias during assessing the scales. Moreover, there were also some confounding bias existing in the study deriving from retrospective study. In the other way, the sample size was small in the study, and it may lead to unstable result.

Response: the limitation of the study had been amendment.

6.The writing of the paper need to improve. There were some sentences hard to understand.For example, page 6,line 161-162, ”This is consistent with a previous study [8] and may be a result of the balance dysfunction in PD correlating with both the dopaminergic and cholinergic system.”. Page 8,line 214-215, “Therefore, the balance function in patients with DBS improved postoperation but was not accurately reflected due to the subjectivity and bias of the balance function score.” and so on.

Response:The paper had been rewritten carefully. Sentences and structure had been amendment logically. 

Minor comments

1.Page 2, line 41-42: Whether the sentence ”Due to the fear of falling, the activities of patients with PD are restricted, which seriously affects their quality of life, and even life safety” was wrong described? That restricting activities of patients with PD would seriously affects life safety?

Response:The sentence had been amendment carefully according to the comments.

2.Page 2, line 45: “different treatment” ought to be “different treatments” . Response:The sentence had been amendment carefully according to the comments.

3.Page 4, line 89-90: “;” should be represented by “,”

Response:The sentence had been amendment carefully according to the comments.

4.Page 4, line 107: whether the “post-operation” represented “4 weeks after operation” or not? 

Response:The sentence had been amendment carefully according to the comments. 

5.Page 5, line 131: “We found no differences in the clinical characteristics of the DBS Group and Medicine Group” would be better to write as “We found no differences in clinical characteristics of the DBS Group and Medicine Group before operation.”

Response:The sentence had been amendment carefully according to the comments.

6.Page6, line 136: “The LEDD preoperation” ought to be “The LEDD in preoperation”

Response:The sentence had been amendment carefully according to the comments.

Reviewer #2: The authors performed a retrospective study of short term effects of STN-DBS stimulation on balance control. The issue is of importance in the field. A great strength of the study is that data from the surgery group are compared to data of a control, best medical treatment group. If correctly analyzed, such data would provide new and relevant information in the field. However, in the present version of the study report, there are major methodological weaknesses in data processing. In addition, the presentation of the methods and results needs to be thoroughly edited. The discussion needs to be focused on the reported data rather than on general considerations about the neuronal networks possibly involved in balance control and balance disorders in Parkinson’s disease patients. Overall, the paper needs serious editing and information needs to be presented in a more structured way. The abstract would also have to be rewritten accordingly.

Response:The abstraction, methods and results had been rewritten thoroughly edited accordingly. 

Major concerns

* The specific variables recorded and analyzed need to be clearly presented and defined. Scoring needs to be explained. It is not clear when the first measure post-surgery was performed? (upon switching on stimulation a few days or weeks after surgery? At three months, after xx weeks of chronic stimulation and adjustments of parameter settings?)

Response:The specific variables recorded and analyzed had been amendment, scoring had been added clearly in the paper.

* It is said in the text that non parametric tests were used, but also that one-way repeated ANOVA were used. Which ANOVA? Friedman? Kruskall-Wallis? GLM? As Tables report F values, it seems that parametric ANOVAs were used. The statistical design to be precisely described. For each dependent variable, there were 4 time-point measures (preop, sometime early post-op, 6 months follow-up and 12 months follow-up) X 2 medication conditions (med off, med on). The most appropriate design (using parametric statistic) would thus be a GLM or ANOVA with a design 2 conditions X 4 time-points (repeated measures for this last factor). If non parametric tests are used, the design is more complex and requires 2 steps.

Response:The Statistical analysis had been amendment accordingly.

* Demographical data cannot be analyzed as the dependent variables of the study. The two groups need to be compared using t tests or a non-parametric equivalent.

Response:The Statistical analysis had been amendment accordingly.

* Data are presented as mean and standard deviation in the Tables, and median and quartiles in the text for some of them (age, PD duration) but not all (LEDD). For the reader, it would be friendlier to have a coherent way of presenting the data. Statistical outcomes for the demographical data need to be reported. One would also want to see the MDS-UPDRS III total score Med off and Med on for the two groups, before surgery (there are given in Tables 2 and 3, but the reader would want to see them with the demographical data, as they contribute to describing the population).

Response:The data presented had been amendment accordingly.

* In Table 1, it is said that males represented 8.5 and 8.4% of the sample; this means that 92% of the patients were females? This would be very unusual and would need to be discussed. In addition, 8.5% of 16 patients are 1.36 persons, and 8.4% of 20 patients are 1.68 persons. This does not make much sense. Examination of Table 1, also shows that disease onset was about 50 years of age in average in the DBS group, and about 45 years of age in the medical 

Response:The data presented had been amendment accordingly.

treatment group. The latter is rather early onset. This should be discussed also.

* Regarding the dependent variables, Tables 2 and 3 report mean and SD, and a single F and p, without explanation of what these Fs and ps refer to: effect of group? Effect of time? What about interactions? Were they significant? If yes post-hoc analyses need to be reported.

Response:The data presented and statistical analysis had been amendment accordingly.

In the text, the outcome of the statistical analyses needs to be precisely reported. “No obvious differences” (page 6, lines 138-139) is not an objective way of reporting. Either there were statistically significant differences, or there were not. Report successively the outcome for each variable.

Response:The sentence had been amendment according to the comments.

Discussion

In its present form, the discussion does not discuss the results of the study. There are many general considerations on balance difficulties in PD, and potentially involved brain regions. The precise outcomes as well as characteristics of the sample and limitations of the study need to be specifically examined and interpreted.

Response:The discussion had been rewritten carefully. Sentences and structure had been amendment logically.

Minor concerns

Response: All the minor concerns had been amendment carefully according to the comments.

Introduction:

* what does “primary” PD means? Idiopathic? (page 2, line 35)

* What do you mean with “increases body swing” (page 2, line 37); please be specific in the description of the sign

* What does “decreases coordination” refer to? Which coordination are you referring to? (page 2, line 37)

* Aim of the study (page 2, line 44-45 “In this study, we retrospectively 45 analyzed patients treated by DBS and medicine to verify the effect of different treatment.”) is too unspecific and should be rephrased. Aim of the study seems to have been to compare the effect of medical treatment vs. STN-DBS on balance control in patients with PD.

Methods:

* Inclusion criteria: please make several sentences or a structured list to make the information legible (page 2 last line to page 3 line 57)

* What was the test and cut-off used to exclude patients with “obvious cognitive dysfunction”? (page 3, line 60); same issue for anxiety and depression.

DBS programming

* Page 4, line 85: contacts, not contactors

* Page 4, line 87: What does it mean to test the stimulation parameters “with attention on the … and drug side effects”? If testing the effect of the stimulation, the unwanted effects are related to stimulation parameters, not to drug.

* Page 4 lines 88-91. “Problems in balance….”. Delete this long sentence. This sentence refers to usual procedures of parameter are considerations on strategies that have been tried to improve balance or gait issues in patients with DBS. You may want to discuss these issues in the discussion, if some of the patients of the present study were actually stimulated in the SNr.

* Page 4 lines 98-103. Again these are general comments regarding management of patients after DBS implantation and is not really relevant to the present report. This paragraph should be deleted, except for the 1st and last sentences. (In addition, FOG can be induced by levodopa, so that increase in levodopa dose is not always the appropriate strategy for clinical management of FOG, especially in patients with dyskinesia.)

* Evaluation of quality of life, motor disability, balance performance and limit of stability (page 4 – lines 104 and following). This whole paragraph needs to be structured. Scales need to be presented (PDQ-39, item 3.12 of MDS-UPDRS III, BBS). Description of the LOS procedure needs editing to be fully understandable (for example, regarding the position of the patients, it would be simpler to state that patients were standing upright, feet at shoulder width, and looking straight ahead at the display. The task needs to be more clearly explained, as well as the variables (what is the “target body swing angle”?). There seems to be a contradiction between performing accurate movements to the target and maximizing movement. “Small limbs” are called “upper limbs” (I suppose).

Thanks for your thoughtful suggestion. I really appreciate that 2 reviewers spend a lot of time and energy on this paper. Your thoughtful comments helped me to present our study structurally and logically. We thought carefully about every word and rewrite the paper.

Kind regards,

Haitao Li, M.M. 

Corresponding author: Xiaoguang Tong

E-mail address: tongxghh@163.com

---

## [Decision Letter · Decision Letter 1]

10 Jul 2020

PONE-D-20-03478R1

Effect of subthalamic nucleus deep brain stimulation (STN-DBS) on balance performance in Parkinson's disease.

PLOS ONE

Dear Dr. Tong,

Thank you for submitting your manuscript to PLOS ONE. After careful consideration, we feel that it has merit but does not fully meet PLOS ONE’s publication criteria as it currently stands. Therefore, we invite you to submit a revised version of the manuscript that addresses the points raised during the review process.

We look forward to receiving your revised manuscript.

Kind regards,

Karsten Witt

Academic Editor

PLOS ONE

Reviewers' comments:

Reviewer's Responses to Questions

**Comments to the Author**

1. If the authors have adequately addressed your comments raised in a previous round of review and you feel that this manuscript is now acceptable for publication, you may indicate that here to bypass the “Comments to the Author” section, enter your conflict of interest statement in the “Confidential to Editor” section, and submit your "Accept" recommendation.

Reviewer #1: All comments have been addressed

Reviewer #2: (No Response)

2. Is the manuscript technically sound, and do the data support the conclusions?

Reviewer #1: Yes

Reviewer #2: Partly

3. Has the statistical analysis been performed appropriately and rigorously? 

Reviewer #1: Yes

Reviewer #2: Yes

4. Have the authors made all data underlying the findings in their manuscript fully available?

Reviewer #1: Yes

Reviewer #2: (No Response)

5. Is the manuscript presented in an intelligible fashion and written in standard English?

Reviewer #1: Yes

Reviewer #2: No

6. Review Comments to the Author

Reviewer #1: Major comments.

1.Specific statistical methods could be described in clinical characteristics of the subjects, instead of nonparametric tests briefly In Page 5, line 128-129.

2.Discussion of the article still lack of attractive. The author could discuss the results more. The detailed characteristics, advantages and clinical significance of this study could be discussed. Moreover, there could have some other mechanisms about the improvement of balance performance in STN-DBS, but not only for the improvement of bradykinesia or rigidity. And limitations of the study could be discussed in detail. In general, the discussion of the article was not well-written.

Minor comments

1.The additional instructions In Page 4-5, line 101-113 were not supposed to be described in the main body of the article.

2. In Page 2, line 36-38, “Balance dysfunction is one of the causes of posture instability, decreases motor coordination, and becomes more and more serious as PD progresses, and has been identified as one of the independent risk factors for falls.” The state was wrong written.

3. In Page 2, line 43, “treatment” could be “treatments”.

4. In Page 6, line 150, “of” could be “or”.

5. In Page 6, line 152-153, what does the author mean by “ different studies have different criteria“?

6. In Page 7, line 180, “ which could inhibit the excitability of the SNr/GPi”. Whether STN-DBS would inhibit the excitability of the GPi or not?

Reviewer #2: The revised version is much improved and the authors strived to answer most of the comments made on the original version.

In addition to serious editing for the English language and syntax, their remains some questions and minor issues.

Abstact

Line 20. “One-way ANOVA” needs to be corrected as the statistical design was repeated measures.

Introduction

Line 45. “In this study we retrospectively analyzed patients to verify the effect of STN-DBS on balance performance.”

The study actually included a group with best medical treatment suggesting that one aim was also to compare STN-DBS and best medical treatment. In the present version, data are presented as parallel analyses in the two groups. There is no direct comparison between the two groups but their alternation in the Tables. However, there is little point in including a group with best medical treatment if not to compare the two procedures. I would therefore suggest to indicate the comparison here and use a statistical design enabling direct comparison.

Methods

Line 56. Third inclusion criterion “MDS-UPDRS III score improvement >30% in acute levodopa drug test”. For the STN group, it should be mentioned that this was expected before surgery. If the criterion also applies after surgery, it needs to be justified and stated, and it should be mentioned whether this in Stim On or Stim Off.

Line 86, BDS programming. “During programming, the limb with severe symptoms was programmed first”. Was it the limb or the side? One supposes it is “with the more severe symptoms”?

Line 96. Post-op first assessment. Please indicate the delay between the time stimulation was switched on and that of the assessment. Was it immediately or within hours? After a day or two?

Line 101. Explain the scoring of the PDQ-39 (whether higher score reflects worse or better quality of life).

Line 104-105. “The evaluation refers to the functional state of the patient who is currently taking anti-Parkinson drug with the best efficacy status.” This is not clear. The MDS-UPDRS is performed both Off and On Med. Under the Off Med condition, the patient has not taken any drug. What does “the best efficacy status” mean?

It is indicated that the same dose of medication was used before and after surgery. Was that the usual morning dose or were the assessments run during a Levodopa challenge (using a greater medication dose)?

Line 120. “The movement is maximized to make the display indicator….” What the authors mean to say is not clear.

Lines 127-131. Statistical analysis.

Demographical data. Only the data at inclusion should to be reported in Table 1 to demonstrate that the two groups did not differ before the study started.

LEDD, MDS-UPDRS III, PDQ-39 are dependent variables with repeated measures and need to be analyzed with a relevant statistical design (Anovas with repeated measures of GLM with repeated measures, as for the other dependent variables).

For all variables, the statistical design should be reported (As mentioned above, I would suggest ANOVA or GLM with 2 Group X 4 Time-points with repeated measures for the last factor). Tables 2 and 3 suggest that two different ANOVAS were run for the DBS group and for the Medicine group. Such design does not enable comparison between the two groups, leading to questioning why a best medical treatment group was included.

Line 132 and below. Results section.

Provide all statistical outcomes. Presumably, with a 2 Group X 4 Time-points design, there should be either an effect of Group, or an effect of Time-point, and an interaction between these two factors. Post-hoc comparisons need to be reported.

Table 3 shows a trend for improvement of MDS-UPDRS III in the Medicine group under the Med On condition. This trend should be discussed.

Table 4 shows trends for improvement of swing angle SD and time in the Medicine group under the Med Off condition. Could this be a learning effect? It should be discussed.

Discussion

Line 154. “In this study patients with H&Y3 stage were included, and certain results were obtained…” “certain results” is too vague.

Line 165-167. “Moreover, due to compensation with a long duration of bradykinesia, rigidity and tremor, the balance dysfunction may be aggravated when rigidity is released by medicine.” Please elaborate the argument.

Line 168. The second sentence is missing its end.

Line 179-180. The mechanisms of action of DBS are still much debated. The authors should be less assertive.

Line 184-185. Regarding balance, Thevathasan et al., 2018, would be a relevant reference.

Line 188. It is not always true that STN-DBS improves camptocormia. In fact, it tends to have no effect (Debû et al., 2018) especially if surgery is performed more than 18 months after camptocormia installation (Schulz-Schaeffer et al., 2015).

Regarding the discussion in general, a number of other studies have previously concluded that STN-DBS does improve balance in the short term (1-3 years) but that balance dysfunction then increases again with disease progression. This should be mentioned in the discussion, and the authors should also elaborate some more on the new information provided by their study, esp. regarding the interest of using objective measures of balance (while clinically, however, the patients get progressively worse and fall more often, which suggests that clinical maneuvers are not so bad to assess postural stability).

7. PLOS authors have the option to publish the peer review history of their article (what does this mean?). If published, this will include your full peer review and any attached files.

Reviewer #1: No

Reviewer #2: No

---

## [Author Response · Author response to Decision Letter 1]

24 Aug 2020

Reviewer #1: Major comments.

1.Specific statistical methods could be described in clinical characteristics of the subjects, instead of nonparametric tests briefly In Page 5, line 128-129.

Response: The statistical methods had been rewrite in detail.

2.Discussion of the article still lack of attractive. The author could discuss the results more. The detailed characteristics, advantages and clinical significance of this study could be discussed. Moreover, there could have some other mechanisms about the improvement of balance performance in STN-DBS, but not only for the improvement of bradykinesia or rigidity. And limitations of the study could be discussed in detail. In general, the discussion of the article was not well-written.

Response: Thanks a lot for your comments. We rewrite the discussion section according to your comments. The clinical significance and limitation of this study had been added into discussion.

Minor comments

1.The additional instructions In Page 4-5, line 101-113 were not supposed to be described in the main body of the article.

Response: The additional instructions had been moved to end of the article.

2. In Page 2, line 36-38, “Balance dysfunction is one of the causes of posture instability, decreases motor coordination, and becomes more and more serious as PD progresses, and has been identified as one of the independent risk factors for falls.” The state was wrong written.

3. In Page 2, line 43, “treatment” could be “treatments”.

Response: It had been amendment according to the comment.

4. In Page 6, line 150, “of” could be “or”.

Response: It had been amendment according to the comment.

5. In Page 6, line 152-153, what does the author mean by “ different studies have different criteria“?

Response:In this sentence, we are trying to say that different including criteria might lead to diverse outcomes in different study. We rewrite the sentence to make it clearly.

6. In Page 7, line 180, “ which could inhibit the excitability of the SNr/GPi”. Whether STN-DBS would inhibit the excitability of the GPi or not?

Response:In this sentence we are trying to say that STN-DBS would inhibit the excitability of the GPi. It had been amendment in the article.

Reviewer #2: The revised version is much improved and the authors strived to answer most of the comments made on the original version.

In addition to serious editing for the English language and syntax, their remains some questions and minor issues.

Thanks a lot for your help in review my article. I really appreciate it.

Abstact

Line 20. “One-way ANOVA” needs to be corrected as the statistical design was repeated measures.

Response: It had been amendment according to the comment.

Introduction

Line 45. “In this study we retrospectively analyzed patients to verify the effect of STN-DBS on balance performance.”

The study actually included a group with best medical treatment suggesting that one aim was also to compare STN-DBS and best medical treatment. In the present version, data are presented as parallel analyses in the two groups. There is no direct comparison between the two groups but their alternation in the Tables. However, there is little point in including a group with best medical treatment if not to compare the two procedures. I would therefore suggest to indicate the comparison here and use a statistical design enabling direct comparison.

Response: thank you for your suggestion in the Statistical analysis.In addition, we analyzed the motor disability and balance function between STN-DBS group and Medicine group. It had been amendment according to the comment.

Methods

Line 56. Third inclusion criterion “MDS-UPDRS III score improvement >30% in acute levodopa drug test”. For the STN group, it should be mentioned that this was expected before surgery. If the criterion also applies after surgery, it needs to be justified and stated, and it should be mentioned whether this in Stim On or Stim Off.

Response: For the STN group, acute levodopa drug test was expected before surgery.It had been amendment according to the comment. 

Line 86, BDS programming. “During programming, the limb with severe symptoms was programmed first”. Was it the limb or the side? One supposes it is “with the more severe symptoms”?

Response: It should be “the limb side with more severe symptoms”.It had been amendment according to the comment. 

Line 96. Post-op first assessment. Please indicate the delay between the time stimulation was switched on and that of the assessment. Was it immediately or within hours? After a day or two?

Response: “the evaluate were underwent in 4 hous after the DBS turned on”.It had been amendment according to the comment.

Line 101. Explain the scoring of the PDQ-39 (whether higher score reflects worse or better quality of life).

Response: “higher score means worse quality of life.”It had been amendment according to the comment.

Line 104-105. “The evaluation refers to the functional state of the patient who is currently taking anti-Parkinson drug with the best efficacy status.” This is not clear. The MDS-UPDRS is performed both Off and On Med. Under the Off Med condition, the patient has not taken any drug. What does “the best efficacy status” mean?

It is indicated that the same dose of medication was used before and after surgery. Was that the usual morning dose or were the assessments run during a Levodopa challenge (using a greater medication dose)? 

Response: This part of article had been rewrite according to the comment.

Line 120. “The movement is maximized to make the display indicator….” What the authors mean to say is not clear.

Response: This part of article had been rewrite according to the comment.

Lines 127-131. Statistical analysis.

Demographical data. Only the data at inclusion should to be reported in Table 1 to demonstrate that the two groups did not differ before the study started.

LEDD, MDS-UPDRS III, PDQ-39 are dependent variables with repeated measures and need to be analyzed with a relevant statistical design (Anovas with repeated measures of GLM with repeated measures, as for the other dependent variables).

For all variables, the statistical design should be reported (As mentioned above, I would suggest ANOVA or GLM with 2 Group X 4 Time-points with repeated measures for the last factor). Tables 2 and 3 suggest that two different ANOVAS were run for the DBS group and for the Medicine group. Such design does not enable comparison between the two groups, leading to questioning why a best medical treatment group was included.

Response: We analyzed the motor disability and balance function between STN-DBS group and Medicine group. It had been amendment according to the comment.

Line 132 and below. Results section.

Provide all statistical outcomes. Presumably, with a 2 Group X 4 Time-points design, there should be either an effect of Group, or an effect of Time-point, and an interaction between these two factors. Post-hoc comparisons need to be reported.

Table 3 shows a trend for improvement of MDS-UPDRS III in the Medicine group under the Med On condition. This trend should be discussed.

Table 4 shows trends for improvement of swing angle SD and time in the Medicine group under the Med Off condition. Could this be a learning effect? It should be discussed.

Response: We analyzed the motor disability and balance function between STN-DBS group and Medicine group. It had been amendment according to the comment.

Discussion

Response: Thank you very much for your comments with this manuscript. I am grateful for your help in reviewing this paper. The discussion section had been rewrite carefully according to the comments.

Line 154. “In this study patients with H&Y3 stage were included, and certain results were obtained…” “certain results” is too vague.

Line 165-167. “Moreover, due to compensation with a long duration of bradykinesia, rigidity and tremor, the balance dysfunction may be aggravated when rigidity is released by medicine.” Please elaborate the argument.

Line 168. The second sentence is missing its end.

Line 179-180. The mechanisms of action of DBS are still much debated. The authors should be less assertive.

Line 184-185. Regarding balance, Thevathasan et al., 2018, would be a relevant reference.

Line 188. It is not always true that STN-DBS improves camptocormia. In fact, it tends to have no effect (Debû et al., 2018) especially if surgery is performed more than 18 months after camptocormia installation (Schulz-Schaeffer et al., 2015).

Regarding the discussion in general, a number of other studies have previously concluded that STN-DBS does improve balance in the short term (1-3 years) but that balance dysfunction then increases again with disease progression. This should be mentioned in the discussion, and the authors should also elaborate some more on the new information provided by their study, esp. regarding the interest of using objective measures of balance (while clinically, however, the patients get progressively worse and fall more often, which suggests that clinical maneuvers are not so bad to assess postural stability).

---

## [Editor Report · Decision Letter 2]

27 Aug 2020

Effect of subthalamic nucleus deep brain stimulation (STN-DBS) on balance performance in Parkinson's disease.

PONE-D-20-03478R2

Dear Dr. Tong,

We’re pleased to inform you that your manuscript has been judged scientifically suitable for publication and will be formally accepted for publication once it meets all outstanding technical requirements.

Kind regards,

Karsten Witt

Academic Editor

PLOS ONE
---

## [Editor Report · Acceptance letter]

2 Sep 2020

PONE-D-20-03478R2 

Effect of subthalamic nucleus deep brain stimulation (STN-DBS) on balance performance in Parkinson's disease. 

Dear Dr. Tong:

I'm pleased to inform you that your manuscript has been deemed suitable for publication in PLOS ONE. Congratulations! Your manuscript is now with our production department. 

Kind regards, 

on behalf of

Dr. Karsten Witt 

Academic Editor

PLOS ONE